# Folding and Stability of Ankyrin Repeats Control Biological Protein Function

**DOI:** 10.3390/biom11060840

**Published:** 2021-06-05

**Authors:** Amit Kumar, Jochen Balbach

**Affiliations:** 1Department of Life Sciences, Faculty of Natural Sciences, Imperial College of Science, Technology and Medicine London, South Kensington, London SW7 2BU, UK; 2Institute of Physics, Biophysics, Martin Luther University Halle–Wittenberg, 06120 Halle, Germany; 3Institute of Technical Biochemistry e.V. and Centre for Structure und Dynamics of Proteins, Martin Luther University Halle–Wittenberg, 06120 Halle, Germany

**Keywords:** ankyrin repeat proteins, protein stability, protein folding, local unfolding, molecular switch, partial unfolding

## Abstract

Ankyrin repeat proteins are found in all three kingdoms of life. Fundamentally, these proteins are involved in protein-protein interaction in order to activate or suppress biological processes. The basic architecture of these proteins comprises repeating modules forming elongated structures. Due to the lack of long-range interactions, a graded stability among the repeats is the generic properties of this protein family determining both protein folding and biological function. Protein folding intermediates were frequently found to be key for the biological functions of repeat proteins. In this review, we discuss most recent findings addressing this close relation for ankyrin repeat proteins including DARPins, Notch receptor ankyrin repeat domain, IκBα inhibitor of NFκB, and CDK inhibitor p19^INK4d^. The role of local folding and unfolding and gradual stability of individual repeats will be discussed during protein folding, protein-protein interactions, and post-translational modifications. The conformational changes of these repeats function as molecular switches for biological regulation, a versatile property for modern drug discovery.

## 1. Introduction

Proteins containing repeating amino acid sequences have drawn great attention in the last decade. Based upon the development in sequencing technology, complete genomes of numerous organisms are available. They revealed that proteins with internal repeat sequences are common in genomic databases, especially those of eukaryotes [1]. Nearly 20 percent of all proteins encoded in the human genome contain repeating units. Next to immunoglobulins, repeat proteins constitute the most abundant natural protein classes [2]. Although repeat proteins are widely distributed in all kingdoms of life, eukaryotic genomes code for more proteins with internal repeats compared to prokaryotic and archaeal genomes. The modular architecture may be key to their evolutionary success. Simple multiplication of existing genetic material enables an organism to evolve protein sequences faster and thus to rapidly adapt to a new environment. Therefore, it is not surprising that repeat proteins are most common in eukaryotes.

Repeat proteins are a fundamentally distinct class of proteins with predominantly α-helical secondary structure when compared with globular proteins. They consist of tandemly repeated modules of 20–50 residues sequence motifs, which stack together to form elongated, non-globular structures. In contrast to globular proteins, repeat proteins are stabilized by the closely spaced residues in the sequence without “long-range” interactions or an extended hydrophobic core. The helices are typically arranged perpendicularly to the elongated linear structure axis and one side of this structure serves as a scaffold for protein-protein interactions. Based on the known structure-function relationship in combination with selection methods such as ribosome or phase display, it is possible to engineer artificial repeat proteins with high specificity for the target proteins [3]. These designed proteins are thermodynamically more stable than their natural counterparts [4]. Because of the lack of disulfide bridges and easy production, artificial repeat proteins are advantageous compared to natural proteins e.g., for engineering binding proteins as well as biotechnological and medical applications [5,6]. The structures and functional properties of repeat proteins have been reviewed by many colleagues [7,8,9,10,11,12,13,14]. Beside the well-known α-helix-based tandem repeats very limited information is available on tandem repeat proteins formed by β hairpins. Recently, MORN repeats from MORN4 were found to function as binding module for Myo3a. The structure of the MORN4/Myo3a complex revealed the formation of an extended single-layered β-sheet structure. It was suggested that β-hairpin-based MORN repeats are generally involved in the protein-protein interactions [15]. Also recently, a new group of RRPNN tandem-repeat proteins has been identified which is a subclass of tetratricopeptide repeats (TPRs). They function as allosteric switches in the quorum-sensing mechanisms of bacteria [16].

In this review we discuss how biological regulation is governed by the graded local thermodynamic stability of ankyrin repeat (AR) protein reflected in protein folding. Four archetypical proteins *viz.* engineered DARPin, the ankyrin repeat domain of the Notch receptor (Nank), the ankyrin domain of the nuclear inhibitor subunit (IκBα) of nuclear factor κB, and the CDK4/CDK6 inhibitor p19^INK4d^ are exemplified here. Ankyrin repeats of Nank and IκBα exist in a partial unfolded state, while these get structured upon binding to the target protein for their biological regulation. In contrast, p19^INK4d^ behaves differently and for regulation its repeats undergo partial unfolding induced by phosphorylation. This local unfolding is important for the subsequent fate of p19^INK4d^ in the human cell cycle. These naturally evolved ankyrin repeat proteins differ from designed proteins from consensus sequences with optimized thermodynamic stability (e.g., DARPins) which show only global unfolding and no low energy folding intermediates. Including homologues example, we will discuss how the modular architecture of ankyrin repeat proteins allows switch on and off biological function by local un- and refolding of individual modules within the evolved scaffold for specific recognition of the target protein. Finally, we will summarize more recent findings of various ankyrin repeats and perspectives for protein engineering and drug discovery [6].

## 2. Structure and Classification

The repeating module of repeat proteins contains secondary structure elements that can fold in a variety of topologies. The linear assembly of repeats results in a simple scaffold, which is dominated by mainly hydrophobic short-range interactions within or with the adjacent repeats [16]. In general, sequentially distant residues (residues of non-adjacent repeats) do not interact with each other. Numerous stabilizing long range interactions, causing complex topologies in globular proteins, are absent in repeat proteins. The lack of these long-range interactions in combination with this simple architecture predestine repeat proteins to an exciting and easy to handle system to study protein folding, stability, effects of post translational modifications (PTMs) including ubiquitination, biological functions, and to facilitate the knowledge for design [13,16]. Figure 1 shows a selection of commonly occurring repeat proteins classified according to their architecture, including the heat repeat (HEAT), armadillo repeat, tetratricopeptide repeat (TPR), leucine-rich variant repeat (LRR), hexapeptide repeat, and ankyrin (ANK) repeat family. 

## 3. Ankyrin Repeats

Among the repeat protein families, the ANK repeat proteins form the largest family, abundantly found in bacteria, archaea, eukaryote, and viral genomes. Since the first discovery of this motif in the yeast cell-cycle regulators Swi6 and Cdc10 [25,26], more than 3000 ANK repeats from ~400 proteins have been identified in the non-redundant protein database and been linked to several human diseases (see recent review [27]). Ankyrin repeats are part of diverse proteins that share the common function to mediate protein-protein interactions for example as membrane bound proteins (e.g., notch membrane receptor), as cytosolic proteins (e.g., cyclin-dependent protein kinase inhibitors of the INK4 family), or proteins in the nucleus (inhibitor subunit (IκBα) of nuclear factor κB) [28,29]. Most proteins contain tandem arrays of helices in 2–7 repeats and up to 33 repeats were found in a single protein [16,29]. The consensus sequence of an ankyrin repeat comprises 33 residues [30], and deviations from this motif typically reduce the thermodynamic stability of the respective repeat [31,32]. Crystal and NMR structures of the ankyrin repeats present e.g., within 53BP2, p16^INK4D^, p19^INK4d^, GABP and IκBα reveal a highly conserved structural motif. The ankyrin repeat is characterized by a pair of antiparallel α-helices that is linked to its neighboring ankyrin repeat via an anti-parallel β-loop, with the first β-strand of the β-loop contributed by the C-terminus of the repeat at position i and the second β-strand contributed by the N-terminus of the ankyrin repeat at position i+l. Tandem ankyrin repeats stack approximately in parallel, so that the helices in one repeat pack against their counterparts in adjacent repeats. The β-strands are roughly perpendicular to the axis of the helices, giving the stack an L-shaped cross-section. There is a left-handed twist to ankyrin repeats and the tandem repeat stack is slightly curved, creating concave and convex faces. Often, the β-loop region can form a continuous β-sheet. One single ankyrin repeat is unable to adopt a folded structure due to its intrinsic instability. Therefore, at least 2 repeat are required as minimum folding units for the formation of a hydrophobic core to overcome chain entropy [33].

The mode of protein-protein interactions that are mediated by ankyrin repeats is highly conserved. In all complexes, contacts between the ankyrin repeats and target proteins involve the β-loop fingers of the ankyrin repeats. Additional contacts can be provided by the surfaces of the inner α-helices of the repeats realized for example in the NF-κB-IκBα complex [34,35] and the CDK6-p19^INK4d^ complex [36,37]. Interestingly, the differing structural features of ankyrin repeats relative to armadillo repeats impose an α-helical conformation on the nuclear localization signal of NFκB when bound to IκBα. The rigidity of tandem ankyrin repeat assemblies may play an important role by modulating conformational changes within the proteins with which they interact. For example, CDK6, NF-κB, and GABP all undergo functionally critical conformational changes upon association with their respective ankyrin repeat partners [34,35,36,38]. 

## 4. General Protein Folding and Stability Aspects

We summarize some basic terms and concepts about protein folding in this paragraph and how these generally apply to ankyrin repeat proteins before discussing examples. The three-dimensional structure of a native protein in its physiological state is typically the conformation with the lowest Gibbs free energy in a given environment and the native conformation of the protein is determined by its amino acid sequence. This dogma going back to early work by Christian Anfinsen [39], got extended by the huge class of intrinsically disordered proteins or domains, which function only in very rare cases in complete absence of structural elements [40]. The mechanism by which the information encoded in the sequence gets translated into the three-dimensional structure is still not fully understood and therefore termed the ‘protein folding problem’. Globular proteins typically fold on a timescale of seconds or less [41]. According to Levinthal’s paradox, proteins cannot fold by a random search of all possible conformations. From the many proposed models and concepts, we will here use the ‘‘new view” or the “energy landscape view’’ of protein folding which gained popularity in the 1990s [42]. A funnel shaped energy landscape with the native structure at its global minimum, guides each molecule from a heterogeneous ensemble of unfolded poly peptide chains of high energy via different microscopic routes down to the bottom of the funnel. If the energy landscape is smooth with only very high energy intermediate states, folding is fast and follows two-state behavior with no detectable intermediates. At rugged energy landscapes with deep local minima, chains easily get trapped and this retardation allows these intermediate states to be detected and studied by biophysical methods [43]. The elongated architecture of ankyrin repeat proteins get locally stabilized by the adjacent repeats rather than contacts to residues distant in sequence. Therefore, folding funnel of one-dimensional architecture have been proposed employing an Ising-like treatment of neighboring interactions between the repeating units [44].

The simplest model for unfolding and refolding involves a single cooperative folding step, in which unfolded (U) and folded (N) states of the protein interconvert: U ⇄ N. This simple mechanism well describes the experimentally observed un- and refolding of many proteins [45], especially when studied under equilibrium conditions [41]. The cooperative nature of this protein folding process is independent of a specific secondary and tertiary structure element attributing to a characteristic fold. The herein discussed ankyrin repeat proteins DARPin, IκBκ, Nank, and p19^INK4d^ deviate from the two-state behavior and even under equilibrium conditions, folding intermediates could be characterized in structural terms (see below). Time-resolved experiments report about the denaturant-dependent un- and refolding rate constants (‘Chevron plots’). They confirm this multi-step folding mechanism of ankyrin repeat proteins including transient states with some of the repeats folded and others not [32,46,47,48]. This allows to characterize these folding intermediates and to locate them on the folding pathways and in energy landscapes of the respective protein. Generally, a nucleation process of individual repeats is assumed for the initial refolding event and apparent two-state folding observed experimentally were discussed according to an Ising-like model of stabilization of these individual repeats by nearest neighbor interactions [4,16,49]. 

## 5. Folding and Function of DARPins

Repeat proteins belong to a class of proteins that can be extended in size while still constituting a contiguous domain. This property gets utilized in biologic and directed evolution. Designed ankyrin repeat proteins (DARPins), for example, have been engineered according to sequence statistics and structural considerations [3]. In a library, designed ankyrin repeat modules have ~67% identity and ~71% similarities to the human purine rich (GA) binding protein (GABP), for example. The continuous elongated hydrophobic core formed by the central repeat modules gets shielded by a capping repeat at the N- and C-terminus, giving a NxC DARPin library (Figure 2), where x is the number of modules [5]. The number of x modules vary from two to five. The consensus design approach helps to build large members exhibiting improved properties, such as very high expression levels, high stability, and solubility. Their thermodynamic stability increases with length and DARPins with more than three internal repeats are very stable against temperature-induced denaturation (thermal midpoints range between 66 °C and above 100 °C) or chemical denaturants (transition midpoint raise up to 7 M of guanidine hydrochloride (GDnHCl)) [4]. Complete denaturation of full consensus ankyrin repeat proteins containing only three repeats require already high temperature and 5 M GdnHCl. Some natural AR proteins show two-state behavior in equilibrium studies, which was observed e.g., for DARPins [30], but kinetic folding intermediates get passed in any case (see below).

All kinetic studies performed with AR proteins so far revealed that the folding mechanism is more complex as expected from equilibrium transitions, with at least one transient folding intermediate. Protein folding kinetics of various DARPins such as NI_x_C have been analyzed, where I represents the full-consensus repeat, the subscript represents the number of identical full consensus repeat modules, and N and C correspond to the N- and C-terminal capping repeats (Figure 2). NI_1_C unfolding and refolding are monophasic, but the unfolding limb in the chevron plot exhibits a curvature [4] representative for a sequential three-state model with a metastable high-energy intermediate [45,50,51]. However, the metastable intermediate is not populated to an extent to cause a second observable folding or unfolding phase. NI_2_C refolding kinetics are monophasic, while the unfolding reaction is best explained by double exponential kinetics. The NI_2_C unfolding limb of the slower unfolding phase fitted to a sequential three-state model, where the unfolding intermediate state is higher populated compared to the NI_1_C DARPin and appears to be formed before reaching the fully unfolded state (on-pathway intermediate) [4]. Therefore, a second kinetic unfolding phase is observed. The kinetics of NI_3_C is more complex: the refolding reaction is monophasic, while unfolding can be divided into three phases [4]. A three-state model is not sufficient to explain such kinetics. With increasing repeat number, the unfolding rates decrease moderately e.g., in TPR proteins [52] but enormously for stable consensus DARPins. The DARPins gained about 11 kcal mol^−1^ in stability per additional repeat compared to 4 kcal mol^−1^ for the TPR representatives. One explanation is that folding of DARPins follows a nucleation process where the assembly of repeats (probably one single consensus repeat) triggers the whole folding pathway. The unfolding is protein length dependent and requires the progressive disruption of condensed folded repeats [4]. 

**Figure 2 biomolecules-11-00840-f002:**
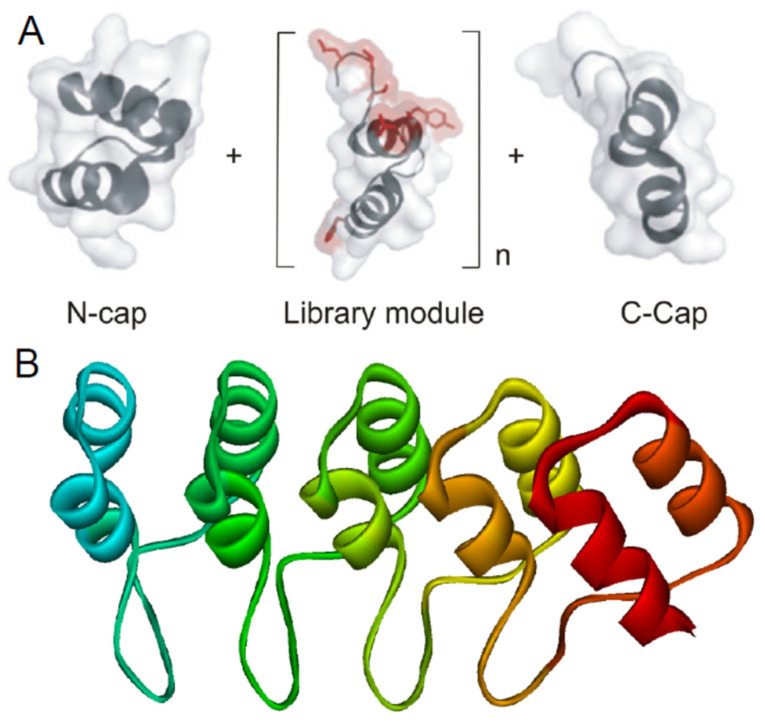
Scheme of the DARPin library design. (**A**) Libraries are composed of capping N- and C-terminus and a varying number of modules (n). The ankyrin repeat module consists of 33 amino acids. The red color highlights the randomized surface for potential target interaction (Figure adopted according to [53]. (**B**) Structure of NI_3_C (PDB ID: 2QYJ) as a representative example rainbow colored from the N- to the C-terminus [5].

DARPins have been used as model proteins for basic research, diagnostic tools, and therapeutic agents [54]. Structures of several DARPins, selected by directed evolution for tight, nano-to-pico molar target binding [55], have been solved by X-ray crystallography with and without bound targets [56,57,58]. DARPins bind to the randomized positions located on the concave molecular surface of the AR scaffold. DARPins or their target do not undergo significant conformational change upon binding; however, some DARPin based inhibitors seem to freeze an inactive conformation of the target protein [57,59]. The stability of the DARPin framework allows introducing broad sequence diversity without affecting the protein structure. The interaction mode is rigid body type and therefore only a minimum of entropy is lost upon binding.

A DARPin based, highly selective caspase-2 inhibitor was produced, showing no cross-inhibitory effect on any of the other caspases [58]. Also kinase binders and inhibitors were obtained which could discriminate between highly similar isoforms [60]. Because of the rigid-body binding, small sequence or conformational differences are sufficient for high specificity of DARPins. For instance, they interact with two subunits of homotrimeric AcrB, a bacterial membrane transporter, while no interaction was observed with the third subunit because this subunit was in a different conformation. This surprising asymmetry allowed structure determination and a more detailed understanding of the drug export mechanism of AcrB. Two consensus AR proteins each from the NI_2_C (E2_5, E2_17), NI_3_C (E3_5, E3_19), and NI_4_C (E4_2, E4_8) AR protein libraries were analyzed in more detail [57,59]. Because of the high stability and affinity, DARPins might mimic the function of natural occurring ankyrin proteins and potentially bind to natural systems. This possibly introduces an immunological tolerance and advantages in drug discovery. One of the attractive therapeutic targets for acute and chronic inflammation is monocyte β_2_-integrin Mac-1. It is crucial for the leukocyte-endothelium interaction. The α_M_ I-domain is an activation-specific epitope of Mac-1 which was targeted using DARPins. This DARPin (F7) specifically binds to activated Mac-1 on mouse and human monocytes which showed reduced leukocyte-endothelium adhesion. Thus, F7 demonstrated therapeutic anti-inflammatory effects in mouse models of sepsis, myocarditis, and ischaemia/reperfusion injury [61]. Another recent application of DARPins was successful in structure biology. Fusion of rigid DARPins fosters protein crystallization, enables recruitment of targets, reduces the size limit for cryo-EM, and allows to investigate spatial restraints in cellular targets [62,63,64]. KRAS represents an undruggable class of biomolecules in downstream signaling pathways in cancer cells. Optimized DARPins could inhibit a KRAS isoform by binding to an allosteric site. This site comprises the KRAS-specific residue histidine 95 at the helix α3/loop 7/helix α4 interface. DARPins directly impair the KRAS/effector interactions and thus the KRAS nucleotide exchange and KRAS dimerization at the plasma membrane [65]. The scaffold engineering, applications, and the success of DARPins as drug platform in clinical studies up to phase III [66], as next-generation therapeutics [62,66,67] and as valuable tools in modern structural biology [62], have been recently reviewed.

## 6. Notch Receptor Ankyrin Repeat Domain (Nank)

Cell-fate decisions are frequently guided by the Notch signaling pathway. This pathway is highly conserved and regulates cell differentiation during development and stem cell homeostasis in the adult organism. Miss regulation such as hypoactive Notch signaling can lead to gross tissue malformations during early development, and hyperactive Notch signaling might lead to T-cell acute lymphoblastic leukemia in children [68,69,70,71]. The Notch receptor is a 300-kDa single-pass transmembrane receptor protein located in the plasma membrane. The Notch intracellular domain (NICD) is composed of the membrane-proximal RAM (RBP-Jk-associated-molecule) region, a seven ankyrin repeat (ANK) domain, a bi-partite nuclear localization sequence (NLS), and a C-terminal PEST degradation motif [72]. The signaling pathway is initiated when a ligand from the DSL (Delta, Serrate, Lag-2, for the mammalian, *D. melanogaster*, and *C. elegans* orthologs, respectively) binds to the extra cellular part of the Notch receptor. This activates a proteolysis near the transmembrane region that releases the NICD from the plasma membrane before translocation into the nucleus. A unique bivalent interaction via NCID engages the CSL transcription for activation with the help of coactivator mastermind (MAML). The RAM and ANK regions were suggested to be critical for activation [73]. Transcription is inactive in vertebrates on individual expression of RAM and ANK [74,75], however, the ANK domain alone activates the pathway in *C. elegans* [76,77]. RAM and ANK binding sites on CSL are the β-trefoil and the C-terminal domain, respectively. The ANK domain only binds at high concentrations and the bulk affinity to NICD:CSL gets assisted by the RAM region [78,79,80]. In higher organisms, the Notch-responsive gene encodes the Notch-regulated ankyrin repeat protein (NRARP). This acts as a negative feedback regulator of Notch responses. The growth of Notch-dependent T cell acute lymphoblastic leukemia (T-ALL) cell lines can be inhibited by NRARP. Here, NRARP with the help of transcription factor RBPJ and NICD binds directly to the core Notch transcriptional activation complex (NTC) without the requirement of Mastermind-like proteins or DNA. The recently solved structure of the NRARP-NICD1-RBPJ-DNA complex revealed that the three ankyrin repeats of NRARP extend the Notch1 ankyrin repeat stack. Thus, NRARP-NICD1-RBPJ complexes require the engagement of RBPJ and NICD1 at the same time [81]. 

The crystal structure of the *Drosophila* Notch ankyrin domain revealed seven ankyrin repeats. The six C-terminal repeats (repeats 2–7) adopt a typical ankyrin repeat fold of two antiparallel α-helices followed by a β-hairpin or a long loop that projects roughly perpendicular to the long axis of the elongated linear array (Figure 3B) [82]. The first N-terminal repeat appears to be distinct from the other six repeats by exhibiting a partly disordered structure. The seventh repeat deviates from the other six by a low average pairwise identity of only 17%, hence it is called putative ankyrin repeat. However, the putative seventh ankyrin repeat greatly increases, as a C-terminal cap, the stability of the *Drosophila* Notch ankyrin domain [83]. 

Urea-induced unfolding transitions of the *Drosophila melanogaster* Notch ankyrin repeat domain (Nank) were monitored by CD spectroscopy (at 222 nm) and fluorescence of the only Trp 157 in ankyrin repeat 5. Both CD and fluorescence detected unfolding curves revealed a cooperative and sigmoidal shape expected from a two-state folding mechanism at equilibrium [83]. In contrast to the above discussed designed AR proteins, here individual repeats contribute differently to the global stability of natural ankyrin domains. This causes the differences in unfolding transition curves for variants of AR domains with deleted repeat modules and Nank nicely illustrates this natural property. Polypeptides denoted here include the N-terminal five, six, and the putative seven ankyrin repeat sequences (Nank1–5*, Nank1–6*, Nank1–7*, and asterisks indicate that cysteines have been replaced with serine). Full-length Nank shows maximal stability. The unfolding midpoint of Nank1–7* at around 2.8 M urea shifts to 2.0 M urea and below if one of the repeats are missing [83]. Addition of the putative seventh C-terminal repeat strongly stabilizes the entire ankyrin domain, which reflects in an increased $\Delta G_{U}^{0}$ (from ~4 kcal/mol to ~8 kcal/mol for Nank1–6) but almost invariant *m*-value. This suggests that the seventh repeat is an integral part of the domain in terms of both structure and stability [83].

The Notch ankyrin domain does not show simple two-state folding kinetics [85]. The refolding and unfolding kinetics are best explained by the sum of two exponential phases. The slow minor refolding phase is limited by a prolyl isomerization. In the minor unfolding phase, an on-pathway intermediate appeared as a lag during fluorescence-detected unfolding. This intermediate is shown to be populated in interrupted refolding experiments allowing its kinetic analysis. The rate constants for the major unfolding/refolding phases each define a V-shaped chevron with non-linear limbs when plotted against the urea concentrations. These two chevron plots yield rate constants for the individual steps in folding and unfolding by fitting a sequential three-state model [85]. Formation of the intermediate state during refolding is rate-limiting and closely matches the major observed refolding phase at low denaturant concentrations. The intermediate appeared to be midway between N and U in folding free energy and denaturant sensitivity, but it exhibited Trp fluorescence properties close to the N state. Although the Notch ankyrin domain has a simple modular architecture, its folding is slow, with the limiting refolding rate constant being seven orders of magnitude smaller than expected from topological predictions [85].

Ankyrin repeats undergo conformational changes in order to control their biological function. The first ankyrin repeat in the Notch receptor domain is significantly disordered in the unbound state [82]. Upon binding to the transcription factor, this repeat becomes ordered and adopts an ankyrin-like fold (Figure 3B) [84,86]. The CSL and Nank interaction creates a binding groove for MAML-1 in a kinked helical conformation. The entire complex is globular containing a cleft separating the DNA binding domain of CSL from Nank, and the overall arrangement of the CSL domains does not significantly change upon complex formation with Nank and MAML-1 (Figure 3A). The consecutive ANK repeats stack together in a curved L-shaped domain creating a concave binding surface. Repeats two to seven of Nank interact with CSL, and this binding induces the curvature [84,86]. The induced folding of the first Nank repeat is likely to be a general phenomenon throughout the Notch family [84] because the primary sequence of the two helices are highly conserved, though residues of the unstructured linker between them are variable. The N-terminal end of Nank is intrinsically more flexible than the C-terminal end [87]. During complex formation, the N-terminal end of the first repeat also approaches the C-terminal end of CSL and the N-terminal end of MAML-1. Alternatively, it was suggested that the first repeat constitutes a recognition element that is induced after formation of this complex in order to recruit additional binding partners [84].

Nank undergoes hydroxylation at one or more asparagines at the side chain C^β^ position [88,89]. Crystallographic analysis of Notch target peptides in complex with FIH (the asparagine hydroxylating enzyme) suggested that the ankyrin domain must be transiently unfolded to be hydroxylated. Hydroxylation is compatible with the native structure [88] and results in an increase in stability [90]. Notch signaling and the hypoxic response mediated by hypoxia-inducible factor-1α are affected by the hydroxylation of Nank [89]. The specificity of FIH towards the respective ankyrin repeat in Nank gets additionally mediated by certain residues proximal to the hydroxylated asparagine [91].

## 7. Ankyrin Repeat Domain of IκBα

The eukaryotic NF-κB nuclear transcription factor [92] family regulates the expression of genes involved in inflammatory and immune responses of the cell, cell growth, and development. Autoimmunity, chronic inflammation, and various cancers have been linked to the inappropriate activation of the NF-κB signaling pathways [93,94,95,96]. However, how these pathways are precisely regulated is often poorly understood. Recently, ankyrin repeat and suppressor of cytokine signaling box containing 1 (ASB1) is shown to enhance the stability of TAB2 and its downstream signaling pathways (NF-κB and MAPK pathways) [97]. A variety of signals, including cytokines, pathogens, injuries, and other stress conditions, lead to activation of NF-κB [98]. NF-κB is a dimer of proteins belonging to the Rel family. All members of the NF-κB protein family, p50 (p105 precursor), RelA (p65), p52 (p100 precursor), RelB, c-Rel, and Relish are monomers, structurally related and share a highly conserved Rel homology region (RHR) made up of 300 amino acids. All the residues necessary for subunit dimerization, sequence specific DNA binding, nuclear localization, and inhibitor binding are located in the RHR. Monomers associate to form transcriptionally competent homo- and heterodimers. Out of these dimers, the p50/p65 heterodimer (prototypical NF-kB) is the most abundant and biologically active one. The activity of the p50/p65 heterodimer is regulated by the members of the inhibitor κB (IκB) family [99,100]. In resting cells, IκB is bound to NF-κB and masks the nuclear localization signal (NLS). This sequesters the NF-κB-IκB complex in the cytoplasm and thus prevents NF-κB from binding to DNA and thus to activate the signaling pathways [101,102,103,104,105].

Members of the IκB family are the canonical IκB proteins (IκBα, IκBβ, and IkBε), NF-κB precursor proteins (p100 and p105), and the nuclear IκBs (IκBζ, Bcl-3, and IκBNS). Structurally, IκB proteins contain an N-terminal signal receiving domain (SRD), a central ANK domain, and a C-terminal proline- glutamate- serine- and threonine-rich (PEST) sequence. IκBα was first discovered as a factor that dissociates preformed NF-κB·DNA complexes in vitro [99,102,105,106]. The structure of IκBα could be solved in complex with NF-κB. The ANK domain consists of six 33-residue ankyrin repeats, each of which contains one β-loop and two antiparallel α-helices. IκBα folds into an elongated, barrel-like structure with a concave and a convex surface (Figure 4) [34,35]. It has been shown that IκBα is only fully folded when bound to NFκB, whereas AR5 and AR6 are distorted in the unbound state. The first, fifth, and sixth ARs of IκBα display a high flexibility and molten globule like character when unbound, and it tends to aggregate at a physiological temperature [107,108]. Under denaturation conditions, the naturally occurring AR domain of human IκBα can be stabilized by consensus mutations Y254L/T257A (YLTA) and C186P/A220P (CPAP) [31]. However, a proteasomal degradation can only be stabilized by YLTA mutations. In these mutations, the gross structure of the protein appears to be similar to wild type but the YLTA and CPAP cause unexpected long-range effects throughout the repeat domains. The C-terminal PEST sequence gets ordered upon YLTA mutations (in the 6th repeat). This phenomenon is not observed in WT or the CPAP mutant. Ordering is proposed to be the underlying mechanism for the extended half-life of YLTA IκBα under in vivo conditions [109]. 

Molten globule states are not characteristic for ankyrin repeat proteins, so it was interesting to study the folding and stability of IκBα. Urea or guanidinium hydrochloride induced unfolding of IκBα do not superimpose when monitored by fluorescence or CD spectroscopy [107]. Folding transitions measured by CD followed a simple two-state folding model. Fluorescence measurement showed an additional non-cooperative transition in the pre-transition phase. The analyzed fluorescence emission arises from a Trp located in AR 6 sensing additional local rearrangements. Additional Trp residues were engineered in AR2, AR4, and AR5. The A133W variant in AR2 for example showed an unequivocally paralleled transition to the CD changes. The ^1^H/^2^H exchange rate reporting about the local stability of the peptide backbone is notably asymmetric for IκBα and its studied variants [107]. Taken together, IκBα displays two folding transitions a non-cooperative conversion at low urea concentration and a major cooperative folding phase upon stronger denaturating conditions. The cooperative transition comprises mainly AR2 and AR3, while the non-cooperative transition arises for AR5 and AR6. Substitutions in the AR2-AR3 caused a significant decrease in the overall stability of the domain. This region corresponds to the repeats that show highest protection against ^1^H/^2^H exchange in the native state, suggesting that the folding appears to be a nucleation process starting at AR2–AR3 and propagating outwards to AR1 and AR4. Folding of AR5 and AR6 occurs in the second folding transition.

Kinetic studies were performed with AR2-AR4 (IκBα 67–206), where Ala 133 was substituted by Trp as fluorescent probe [110]. Urea induced unfolding of IκBα 67–206 fit to single exponential kinetics, whereas refolding analysis required two to three exponential terms. The Chevron plot of this truncated IκBα reveals two distinct regions of unfolding between 0 M and 1.5 M urea and above 5.5 M urea (‘roll-over’). The unfolding phase connects well with one of the three refolding phases at about 2.8 M urea. These main refolding phases showed strong denaturant dependence and accounted for 68–85% of the total amplitude. The slowest refolding phase was insensitive to denaturant concentration and accounted for 18% of total amplitude, with a rate of 9.3 × 10^−3^ s^−1^, up to urea concentrations of 2.5 M, beyond which the phase was not observed. Interrupted unfolding experiments show that this phase is due to a slow isomerization of one or more non-native Xaa-Pro conformations in the unfolded state. The third phase accounted for 15% and appears only below 2 M urea. A linear four-state model D ⇄ I_1_ ⇄ I_2_ ⇄ N was required to fit to the data [110] with two on-pathway, high-energy intermediates [50,111]. This model describes two intermediates, which do not populate to a detectable amount at equilibrium and are of higher free energy compared to the unfolded or folded state. Folding kinetics of the extended AR domain IκBα 67–287 was found to be similar to the four repeat IκBα 67–206. This suggested that the presence of AR5 and AR6 does not affect the main folding route of the IκBα AR domain [110]. 

IκBs interacts with NF-κB via its ANK domain (Figure 4). The canonical IκB subfamily has a preference for NF-κB dimers containing a p65 or c-Rel subunit, whereas the nuclear IκBs prefer p50 or p52 homodimers [112]. The two classes follow a common mechanism of interaction [34,35,113], where an NF-κB dimer binds to one molecule of IκB . The structure of the p50·p65·IκBα complex (Figure 4) revealed that ankyrin repeats 1 and 2 bind the NLS of p65, repeats 4 and 6 contact p50 at the interface of the paired dimerization domains, and repeats 5 and 6 contact the dimerization domain of p65 [34,35]. To activate NF-κB, inhibiting IκBα gets phosphorylated at the N-terminus before the first AR, which induces degradation via the ubiquitin-proteasome pathway [114]. Uncomplexed NF-κB binds to κB DNA after translocation to the nucleus to initiate transcription of downstream genes. The latter transcripts include IκBα, which binds again to NF-κB and thus strips it from the DNA. Finally, the NF-κB·IκBα complex returns back to the cytoplasm [115]. During termination of NF-κB signaling via DNA stripping, the p65 subunit undergoes a conformational change in the presence of IκBα and adopts a closed conformation. The NTD of p65 shifts by ~38 Å toward its dimerization domain and rotates ~180° about its axis. These conformational changes are important for allosteric inhibition of DNA binding of NF-κB. IκBα interacts directly to the DNA-binding residues of NF-κB, where ankyrin repeat 6 and the C-terminal PEST residues of IκBα interact with the RHR-NTD and interfere with DNA binding [116]. Additionally, the six ankyrin repeats of IκBα show structural transitions upon binding to NF-κB. In solution, repeats 2–4 are structured, but repeats 1, 5, and 6 are significantly disordered [107]. Upon binding to NF-κB, repeats 5 and 6 become structured, although repeat 1 appears to retain substantial disorder. The central array repeat 3 appears to become less ordered upon binding to NF-κB [34,35,117]. It has been shown that using consensus sequences to stabilize the ankyrin repeat fold led to decreased affinities of IκBα and NF-κB [118,119]. This suggested that the structural plasticity and rearrangements of AR can be essential for complex formation and biological regulation.

## 8. CDK4/6 Inhibitor p19^INK4d^

Cyclin-dependent kinases (CDKs) control the eukaryotic cell cycle [120,121]. CDK4/cyclin D, CDK6/cyclin D, CDK2/cyclin E, and CDK2/cyclin A regulate G1 progression and entry into the S phase of the cell cycle [122,123]. In the G1 phase, retinoblastoma tumor suppressor protein (pRb) binds to the eukaryotic transcription factor E2F and thus prevents E2F mediated gene expression. Upon entry into the S phase, CDK4 and CDK6 phosphorylate pRb, disrupt the pRb-E2F complex and thereby inhibit its growth-suppressive function. This triggers activation of E2F-dependent transcription that is necessary for completion of G1 and entering the S-phase of the cell cycle [124,125]. CDK activity is further regulated by CDK inhibitors (CDKIs) which help in inducing cell-cycle arrest in response to different signals [126,127]. Two different classes of CDKI have been identified. Members of the Cip/Kip family include p21^Cip1,WAF−1^, p27^Kip1^, and p57^Kip2^. These inhibit all G1- and S-phase CDKs and are important in p53- and TGF-β mediated cell-cycle arrest [126]. The second class are members of the inhibitor of kinase 4 (INK4) family, including p16^INK4a^ , p15^INK4b^ , p18^INK4c^, and p19^INK4d^ [128,129,130,131,132]. These are specific for CDK4 and CDK6 [132] and can bind in presence or absence of cyclin D. Changes in the expression level or mutations in cyclin D1, CDK4, CDKI and pRB are strongly implicated in cancer [123,133,134,135,136].

The four members of the INK4 family structurally share a similar protein fold. P16^INK4a^ and p15^INK4b^ comprise four and p18^INK4c^ and p19^INK4d^ five ankyrin repeats. Characteristic for all members is helix two of the second AR, which consists of just one helical turn compared to the canonical AR fold [137,138,139,140]. Although the INK4 members appear structurally redundant and all serve as inhibitors of CDKs, there are several non-overlapping features. Few of the members participate in basic processes such as DNA repair, terminal differentiation, and cellular aging or senescence and their unique expression patterns dependent on cell und tissue type as well as the differentiation stage [95,141,142,143]. Surprisingly, p16^INK4a^, p15^INK4b^ and p18^INK4c^ exhibit a lower thermodynamic stability when tested by urea and GdmCl transitions under in vitro conditions compared to p19^INK4d^ [46,144]. In cell lines, the half-life of p16^INK4a^ , p15^INK4b^ and p18^INK4c^ vary from 4–6 h as compared to the p19^INK4d^ which was found to be rapidly degraded with protein half-life of 20–30 min. Of the four members, it is only p19^INK4d^ whose periodic oscillation is determined by the ubiquitin/proteasome-dependent mechanism [142,145]. Post-translational modifications, such as phosphorylation, were not detected for p15^INK4b^ and p16^INK4a^, but for p18^INK4c^ and p19^INK4d^. The latter showed prominent single and double phosphorylation of Ser66 and Ser 76 in U-2-OS cell lines [143]. When the cells are in genotoxic stress such as incubation with Aβ amyloids or irradiated with ultraviolet light, Ser76 and Thr141 are the p19^INK4d^ phosphorylation sites [143,146].

p19^INK4d^ contains ten sequentially arranged helices forming the five ankyrin repeats AR1–5 (Figure 5A). Each repeat consists of an extended strand followed by a helix–loop–helix motif and another extended strand. A series of β-turns between the N- and C-termini links the consecutive repeats. The stacking of antiparallel helices in each repeat creates the elongated L-shaped structure of p19^INK4d^. The helical bundles form the long arm of the L, whereas the extended strands and β-turns form its base [36,37,138,139,147]. p19^INK4d^ binds through its concave face to one side of the active site of CDK6 at its cleft between the N- and C-terminal domains (Figure 5D).

Wild type p19^INK4d^ neither contains Tyr nor Trp [36]. Therefore, urea induced unfolding transitions were monitored by far-UV CD or intrinsic phenylalanine fluorescence and 2D ^1^H-^15^N-NMR spectroscopy. The CD spectrum shows distinct minima at 207 nm and 222 nm characteristic of α-helical backbone conformation. Denaturation of the protein with high concentrations of urea results in a loss of the minimum at 222 nm. The transition was found to be cooperative with a midpoint of 2.9 M urea at 15 °C. A similar transition was observed when monitored by intrinsic phenylalanine fluorescence, rarely accessible for other proteins. The CD and fluorescence transitions coincide and suggested a cooperative two-state folding mechanism for equilibrium unfolding [144]. The NMR detected transition revealed an intermediate at equilibrium with a maximum population of about 30% at about 3 M urea. This observation was later confirmed by a global analysis of fluorescence detected folding kinetics of p19^INK4d^ [46]. These kinetics were recorded by the F86W variant, which is still able to bind to CDK6, followed by stopped-flow fluorescence spectroscopy. Unfolding kinetics are a biphasic process during which the fluorescence increases strongly and reaches a maximum at about 1.3 s before strongly decreasing to a final value that is lower than the fluorescence of the folded protein. This maximum corresponds to a hyper fluorescent folding intermediate. The refolding reaction is also biphasic with increasing fluorescence intensity for both phases. The analysis of the urea dependent un- and refolding rate constants of both phases resulted in a sequential three state model U ⇄ I ⇄ N and the determination of all four intrinsic folding rates *k*_UI_, *k*_IU_, *k*_IN_, and *k*_NI_ [46]. Unfolding and refolding were also measured by stopped-flow far UV-CD spectroscopy and the rate constants coincided with those measured by Trp fluorescence. In addition, p19^INK4d^ shows a very slow refolding reaction that is limited by prolyl isomerization with a urea-independent rate constant of 0.02 s^−1^ [46]. This prolyl cis/trans isomerization [148] can be accelerated tenfold [46] in the presence of equimolar prolyl isomerase SlyD(1–165) from *Escherichia coli* [149,150].

The local stability of individual repeats of p19^INK4d^ were followed by NMR detected ^1^H/^2^H amide proton exchange. Backbone amides in AR 3 and 4 are most protected against exchange with protection factors up to 2 × 10^5^. In repeat 5 protection, on average, was tenfold reduced, and, in repeats 1 and 2, about 100-fold lower compared to AR3–4 [46]. This indicates that AR1 and 2 undergo ‘local breathing’ allowing faster ^1^H/^2^H exchange compared to the scaffold part formed by AR3–4 and possibly 5. Taken together the folding of p19^INK4d^ is a sequential two-step reaction via a hyperfluorescent on-pathway intermediate. This intermediate is observed under all conditions during unfolding, refolding, and at equilibrium. The N ⇄ I transition is much faster compared to the I ⇄ U transition. Under equilibrium conditions the intermediate populate close to the transition midpoint. NMR and truncated variants analyses showed that the N-terminal repeats 1 and 2 are not folded, whereas the C-terminal repeats AR3–5 are already folded in the on-pathway intermediate [46]. During refolding of p19^INK4d^ repeats, AR3–5 first form the stable scaffold for the subsequent assembly of repeats AR1 and AR2. The graded stability and the facile unfolding of repeats 1 and 2 is a prerequisite for the downregulation of the inhibitory activity of p19^INK4d^ during the cell-cycle (see below).

A graded stability is a common property of natural ankyrin repeat proteins and was already discussed above for Nank and IκBα. Another example is the protein tANK from the, compared to human, evolutionary much older archaeon *Thermoplasma volcanium*. It shows < 25% sequence identity but a homologues structure to p19^INK4d^, also comprising 5 AR [32]. The GdmCl induced unfolding of tANK showed a two-step transition under fluorescence detection and NMR revealed that residues of AR 1 and 2 unfold during the first transition at 1.6 M GdmCl and residues of AR 3–5 during the second transition at 2.8 M denaturant. Because of this clear separation of two cooperative unfolding transitions, the intermediate state with only AR3–5 folded populates at 2.2 M GdmCl up to 90% under equilibrium [32]. In both proteins, p19^INK4d^ and tANK, the primary sequences of AR3–5 are closer to the 33 residue consensus motif of ankyrin repeats resulting in a high thermodynamic stability. Kinetic folding experiments revealed again a sequential two-step folding mechanism U ⇄ I ⇄ N where the intermediate forms during the slower phase of refolding [32].

In parallel to Nank and IκBα described above, the graded stability of the different AR of p19^INK4d^ has also direct functional implications. P19^INK4d^ facilitates binding through its concave face to CDK6 mainly by AR1 and 2 (Figure 5). AR 2 deviates most from the AR consensus sequence [138,139,146]. AR 1 and 2 forms after scaffolding AR 1–3 during folding and show the lowest stability in amide proton exchange experiments. This low stability allows the cell to regulate the p19^INK4d^ mediated inhibition of CDK6 by posttranslational modifications (see below). On the CDK6 side, the interaction with p19^INK4d^ occurs mainly through the β-sheet in the N-terminal domain, loop L7 linking the N- and C-terminal domains, and helix α2 in the C-terminal domain [36]. The ternary complexes of p19^INK4d^ with cyclin D/CDK6 show that INK4 protein binding does not obstruct the cyclin binding site [128,131,139].

Many cellular processes are controlled by covalent protein modification including phosphorylation [37,151,152]. To mimic phosphorylation of p19^INK4d^, Ser66 and Ser76 were substituted by glutamic acid introducing a negative charge and to approximate the function. Serine 76 substitution strongly reduced the stability of the native state but not of the on-pathway intermediate. A detailed structural analysis revealed that the residues of AR 3–5 are still forming a native conformation, whereas AR 1–2 are partially unfolded at elevated temperature of 37 °C [46]. MD simulations highlighted the molecular origin of the reduced stability. Phosphorylation of Ser76 strongly destabilizes the interface between AR 2 and AR 3, mainly by disturbing the hydrogen bonding network of adjacent residues. This decouples AR 1–2 from the stabilizing repeats AR 3–5. Ubiquitination was only observed for p19^INK4d^ variants with two substitutions with glutamic acid at positions Ser66 and Ser76. This already implied that a double phosphorylation is required to induce degradation of the CDK4/6 inhibitor p19^INK4d^ via the proteasome pathway.

The entire cellular regulation of the CDK4/6 inhibitor p19^INK4d^ by phosphorylation could be disclosed by using cell lysate [37] in order to get close to the biological system (Figure 5). Phosphorylation of p19^INK4d^ at Ser66 by the kinase p38 and Ser76 by CDK1 is a sequential process. The Ser66 modification destabilized p19^INK4d^ via repulsive electrostatic interactions of the phosphate moiety and the negative charge of the net dipole moments at the C-terminal ends of helices 4 and 6 in AR2 and AR3, respectively (Figure 5B). This destabilization was not sufficient to locally unfold p19^INK4d^ but the prerequisite for the second modification of Ser76. The latter was strictly cell cycle dependent and induced a unique type of local unfolding of the three N–terminal ankyrin repeats 1–3. This conformational switch dissociates the CDK6–p19^INK4d^ complex and exposes Lys62 to be targeted for ubiquitination and subsequent degradation. The in cell biophysics of p19^INK4d^ phosphorylation was later verified in vitro by chemical synthesis [153]. Compared to above discussed IκBα, where phosphorylation of S32 und S36 in the unstructured N-terminus before the first AR targets the inhibitor to the ubiquitin-proteasome pathway [114], this PTM of p19^INK4d^ first unfolds AR 1–3 before degradation. Most mutations reported so far in the INK4 genes have resulted in loss of function due to incorrectly folded and/or insoluble protein variants. These recent findings demonstrate how phosphorylation induced conformational changes of local unfolding of ARs control the fate of p19^INK4d^ during the human cell cycle and highlight that an untimely event could also result in a deregulated cell cycle [37]. This makes p19^INK4d^ a promising drug target as next generation CDK inhibitors [154].

Gankyrin is another cell-cycle related ankyrin repeat protein comprising seven repeats [23,155,156]. Oncogenic gankyrin binds, but does not inhibit, CDK4 in the same mode as members of the INK4 family. Additionally, it binds to S6 ATPase of the 26S proteasome and is involved in the degradation pathway of pRb [157]. A mutation of L62H/I79D led to considerable destabilization of the repeat fold at body temperature and rearrangements of the loops and (T/S)PLH regions of the N-terminal repeats [158]. The mutation at I79D inhibits the kinase activity of CDK4, similar to p16^INK4a^. Further structural and biophysical analyses suggest that the substitution of Ile79 with Asp leads to local conformational changes in loops I–III of gankyrin. This retains its binding capacity but now as inhibitor of CDK4 [158]. Differences in the stability of the seven ARs determine also the complex protein folding landscape of gankyrin [159]. Refolding and unfolding follow two different pathways. During unfolding kinetics, the N-terminal ARs denature first and at least one intermediate is formed. Upon refolding, the order of events is reversed, and the ARs form stepwise from the N- to the C-terminus.

## 9. Recent Findings of Various AR Proteins

This review cannot cover folding, stability, and related functional properties of all ankyrin repeats of this huge protein family. Therefore, four archetypical examples were selected for a detailed summary and we finish with few interesting recent findings from other AR containing proteins. The ribonuclease activity of RNase L can be regulated by oncolytic viruses and the kinase inhibitor sunitinib which has synergistic effects in anticancer therapy. Structural analyses showed that sunitinib binds to the ATP-binding pocket of RNase L. An unusual flipped-orientation was observed in the structure because sunitinib affected the binding mode of the αA helix linking the ankyrin repeat domain and the PK domain. Furthermore, the dimer conformation of RNase L destabilized upon binding of sunitinib that allosterically inhibited its ribonuclease activity [160]. In patients suffering from psychiatric disorders, Shank1/2/3 at excitatory synapses are frequently mutated. An interaction of GTP bound Ras and Rap1 with the Shank N-terminal domain and the ankyrin repeat domain tandem (NTD-ANK) is well established. Recently, it has been shown that NTD and ANK together led to the formation of a so far unknown binding site. Additionally, GTP hydrolysis of GTP-loaded Rap1 slows down upon binding to Shank3. Thus, it has been concluded that the signaling of the Ras family proteins can be modulated by Shank3 via its NTD-ANK binding by stabilizing the GTP-bound form of the enzymes [161]. The transient receptor potential (TRP) channel TRPV4 acts as mechanosensor and osmosensor for calcium in skeletal muscles. Fundamentally, ATP binding within the ankyrin repeat domain (ARD) of TRPV4 is the underlying mechanism to cause osteoarthropathy related diseases. Non-conventionally, mutations in the ARD can reduce the TRPV4 activity. However, the underlying mechanism was not known. Recently, significant alteration on the conformation upon point mutation in the ankyrin repeat domain explained the propagation from the remote mutation site to the altered ATP binding site of TRPV4-ARD [162]. The chloroplast signal recognition particle (cpSRP43) is essential for biogenesis of the light harvesting chlorophyll-binding proteins (LHCP). In order to increase the efficiency of capture and solubilizing LHCPs, cpSRP43 is activated by a stromal factor, cpSRP54. Now, it has been shown that the substrate binding domain of cpSRP43 undergoes a disorder-to-order transition of the ankyrin repeat motifs that drives its activation. Thus, these studies have revealed how the chaperone activity is regulated by the conformational plasticity of cpSRP43 and suggests a general mechanism by which ATP-independent chaperones can be regulated [163].

## 10. Conclusions

The modular structure of repeat proteins contains no direct contacts between distant parts of the protein chain. Unlike globular proteins, repeat proteins have a low contact order [164]. Protein folding transitions of globular proteins are often highly cooperative. From a topological point of view, it is surprising that many repeat proteins show cooperative equilibrium unfolding transitions. If the different ARs are of comparable stability developed during biological or directed evolution, one-dimensional folding funnel guide the polypeptide chain towards the native state. Through local destabilizations and variations in stability on a medium length scale, it is evident that partly folded states, in which some repeats are ordered and others are disordered, can populate. This principle is conserved in all kingdoms of life, has been also reviewed elsewhere [165], and for four examples of ankyrin repeat proteins and their homologs most recent findings have been discussed in detail in this review. The graded stability might be the result of a deviation from the highly stable consensus sequence, a post-translational modification, a missing binding partner, or a combination of these causing a local unfolding and thus loss of function. In vitro protein folding experiments can disclose these ‘predetermined breaking points’, because they also determine the folding mechanism allowing a more rational investigation of the in vivo system. The less stable ankyrin repeats are molecular ‘switches’, where a small modification or local change in the environment is enough to tip the balance from the folded to the unfolded state or vice versa, and the cell uses this property to control function. The concept of molecular switches has been also recently discussed for other repeat proteins [16]. The more stable repeats built the scaffold parts of the proteins as platform for the reversible conformational change to take place. Extremely stable AR proteins can be achieved by protein design and engineering but might be counterproductive in a biological system, which has to switch on and off protein function, the latter often accomplished by proteasomal degradation. The graded and tunable stability of individual ARs is a versatile property for drug discovery [166]. As future perspective, it is worth to look at ARs, which deviate from the consensus sequence and pure scaffolding properties. During design, a combination of highly and less stable AR might accommodate the conformation plasticity found in biological systems. AR, which get destabilized by PTMs or which form late during folding, might be promising candidates targeted by drug molecules to intervene with their function, and biophysical studies can disclose such candidates. A comprehensive understanding of cellular protein function also includes its controlled degradation. For IκBα and p19^INK4d^ we summarized here the current state, and more AR examples can be found in the recent review [13]. Thus, one of the future challenges includes a better understanding of the control of the various cellular functions of AR proteins and their degradation.

## Figures and Tables

**Figure 1 biomolecules-11-00840-f001:**
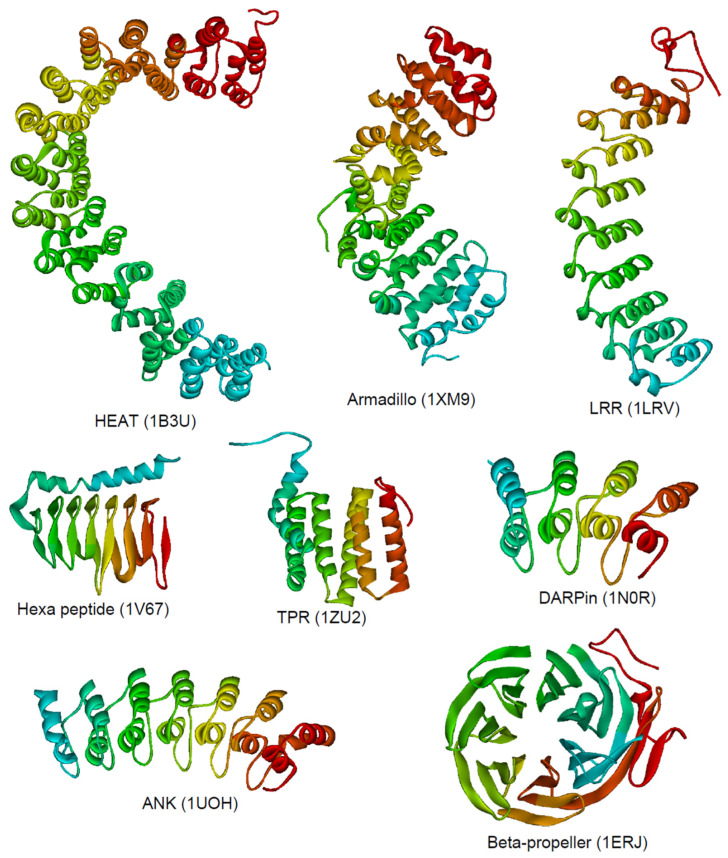
Structural architecture of various repeat proteins. The backbone is depicted from the N-terminus (red) to the C-terminus (cyan) for the HEAT repeat protein phosphatase 2A PR65/A subunit [17], the armadillo repeat domain of plakophilin 1 [18], an leucine-rich repeat variant [19], the hexa peptide repeat gamma-class carbonic anhydrase [20], the TPR repeat domain of TOM20 [21], DARPin [22], the ankyrin repeat protein gankyrin [23], and the beta propeller domain of Tup1 [24]. The PDB ID is given below the structure.

**Figure 3 biomolecules-11-00840-f003:**
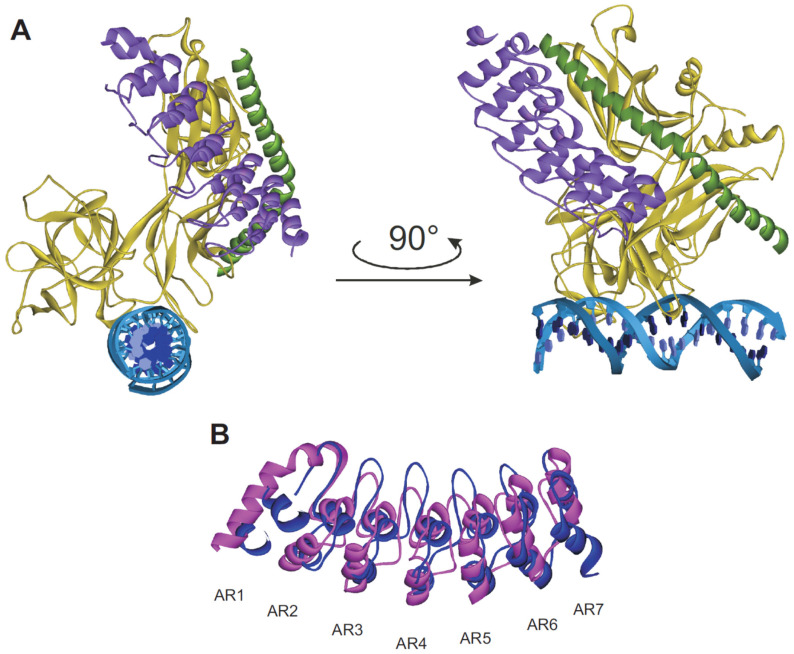
Proposed functional model of Nank. (**A**) Overall structure of the MAML-1:ANK:CSL:DNA complex (PDB ID: 2F8X) in ribbon representation. The ankyrin domain is purple colored, the MAML-1 polypeptide is colored as dark green and the RHR-N, β-trefoil, and RHR-C domains of CSL are colored gold. The two DNA strands are colored blue and cyan [84]. (**B**) Superposition of Nank showing the conformational changes induced in the AR1 upon complex formation. Free Nank is depicted in magenta (PDB ID: 1OT8) [82] and Nank in the complexed state in blue (PDB ID: 2F8X) [84].

**Figure 4 biomolecules-11-00840-f004:**
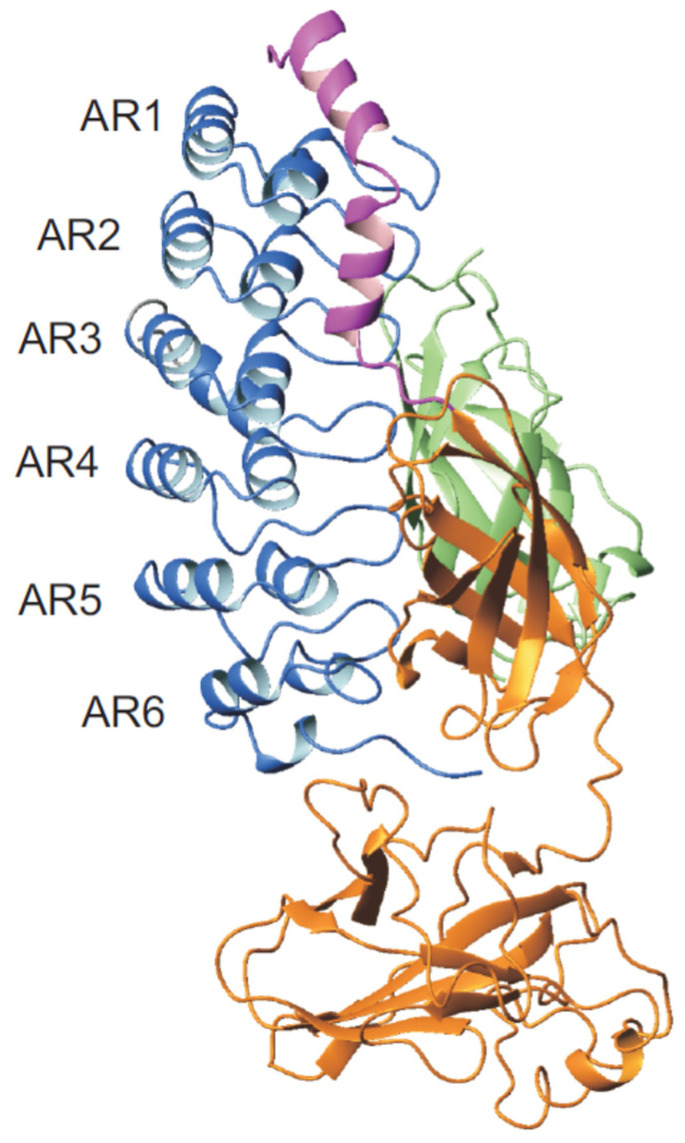
Structure representation of the IκBα ankyrin domain in complex with NF-κB. The IκBα ankyrin domain, composed of six repeats, is illustrated in blue, p50 in green, p65 in orange, and the p65 NLS in magenta (PDB ID: 1NFI) [35].

**Figure 5 biomolecules-11-00840-f005:**
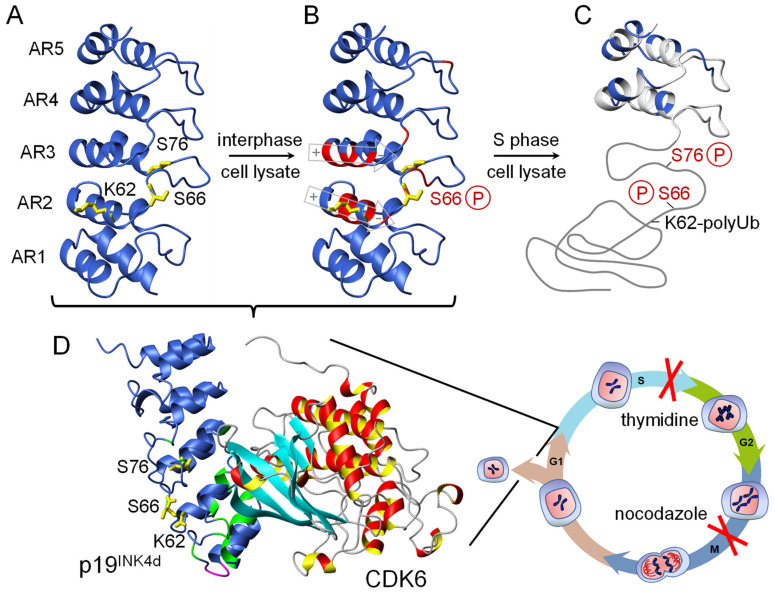
Cell cycle-dependent conformational changes of ankyrin repeat protein p19^INK4d^ [37]. Phosphorylation at position Ser66 by p38 and at position Ser76 by CDK1 locally unfolds ankyrin repeats 1–3 to release CDK6, which is active during cell cycle progression from the G_1_ to S phase. Both kinases could only be identified by arresting the cell-cycle after the S phase and the G_2_ phase. Poly-ubiquitination (polyUb) as the signal for degradation occurs only in the double-phosphorylated state. (**A**) Residues K62, S66, and S76 are indicated in yellow. (**B**) Residues with significantly affected backbone NMR chemical shifts upon phosphorylation of S66 are indicated in red. The macroscopic helix dipole moment of helix 4 and 6 is depicted in gray. (**C**) Only repeats 4 and 5 remain folded after the second phosphorylation of S76, indicated by native backbone chemical shifts (blue). (**D**) Crystal structure of the CDK6/p19 complex (1BLX.pdb). Green represents residues, which in solution show NMR chemical shift changes upon CDK6 binding.

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
