# Peer review of "Folding and Stability of Ankyrin Repeats Control Biological Protein Function"

_biomolecules, 2021, doi:10.3390/biom11060840_

Round 1
Reviewer 1 Report
The review is related to the analysis of structural features and protein stability of ankyrin repeats, tandemly repeated modules of sequence motifs forming nonglobular elongated structures. It is a nice presentation of the current state of the study. I will recommend the publication of this work in Biomolecules.
I have several points which in my opinion not well presented in the review.
1) Authors are comparing containing ankyrin repeats in several proteins, DARPins, Notch receptors, and others. It is not clear from the review how conserved a single module within and between these structures. What size of a single module? Is 33-residue repeat universal for different proteins?
2) Is it any chance that the ankyrin module is a stable structure by itself and what is the mechanism of its stability. What is the 'critical' length of repeats to create a stable structure?
Author Response
The review is related to the analysis of structural features and protein stability of ankyrin repeats, tandemly repeated modules of sequence motifs forming nonglobular elongated structures. It is a nice presentation of the current state of the study. I will recommend the publication of this work in Biomolecules.
I have several points which in my opinion not well presented in the review.
1) Authors are comparing containing ankyrin repeats in several proteins, DARPins, Notch receptors, and others. It is not clear from the review how conserved a single module within and between these structures. What size of a single module? Is 33-residue repeat universal for different proteins?
Answer: Ankyrin repeats share a 33 residue consensus motif, which is well conserved and all examples given in the review share it. We mention this now more explicitly when introducing the ankyrin repeats on page 4 and when presenting the stability properties of the different examples e.g. on page 16, to better support our statement to the consensus motif in the conclusions.
2) Is it any chance that the ankyrin module is a stable structure by itself and what is the mechanism of its stability. What is the 'critical' length of repeats to create a stable structure?
Answer: This module property has been tested experimentally. We mentioned this on page 4 and gave the corresponding ref. [16] there. A single repeat is not stable. The critical length is at least two repeats that can form a hydrophobic core to overcome chain entropiy during folding.
Reviewer 2 Report
This paper reviews the ankyrin repeats control biological protein functions. The whole paper is well organized, and literature is well cited. This topic is significant for biomolecular studies. I have only one minor comments:
Comments:
- It’s better to describe in more details what are the challenges and important tasks in the future research in this area. This helps the readers, especially who are new to this field to think about what to do in the future.
Author Response
This paper reviews the ankyrin repeats control biological protein functions. The whole paper is well organized, and literature is well cited. This topic is significant for biomolecular studies. I have only one minor comments:
Comments:
1) It’s better to describe in more details what are the challenges and important tasks in the future research in this area. This helps the readers, especially who are new to this field to think about what to do in the future.
Answer: We updated the entire review with very recent findings not only for the archetypical AR proteins discussed in detail but also related ones, which should be stimulating for future ideas. Additionally, we give now some future directions at the very end of the conclusions.
Reviewer 3 Report
The authors reviewed the folding and stability of ankyrin repeat domain and its biological function. However, there are serious concerns on this manuscript.
Review articles in scientific journal should summarize and discuss the contents of recently published papers. However, this review seems to describe the research history as it mainly deals with contents published long ago.
According to Pubmed, from 2017 to the present, more than several hundred articles related to Ankyrin domain have been published. Nevertheless, the author was citing only 4~5 articles from 2017. More than 500 papers have been published from 2019 to the present, but they have not been cited at all. Reviewing papers that were reported long ago does not provide useful information to readers.
The author should rewrite the manuscript, focusing on the recently published papers.
Author Response
The authors reviewed the folding and stability of ankyrin repeat domain and its biological function. However, there are serious concerns on this manuscript.
Review articles in scientific journal should summarize and discuss the contents of recently published papers. However, this review seems to describe the research history as it mainly deals with contents published long ago.
According to Pubmed, from 2017 to the present, more than several hundred articles related to Ankyrin domain have been published. Nevertheless, the author was citing only 4~5 articles from 2017. More than 500 papers have been published from 2019 to the present, but they have not been cited at all. Reviewing papers that were reported long ago does not provide useful information to readers.
The author should rewrite the manuscript, focusing on the recently published papers.
Answer: The manuscript was prepared for the special issue of Biomolecules dedicated to Chris Dobson and his lifetime achievements. This might justify the ‘research history’ style. Following the advice of the reviewer, we rewrote the manuscript by adding recently published papers to each of the four archetypical examples we reviewed in great detail plus a new paragraph summarizing very recent findings about molecular details of the biological function of ankyrin repeat proteins. All together, we review now 20 papers published between 2019 and 2021.
Round 2
Reviewer 3 Report
If this paper is for dedication, it will be OK for publication.
The paper has been improved by updating more recent works